# Cord Blood Plasma and Placental Mesenchymal Stem Cells-Derived Exosomes Increase Ex Vivo Expansion of Human Cord Blood Hematopoietic Stem Cells While Maintaining Their Stemness

**DOI:** 10.3390/cells12020250

**Published:** 2023-01-07

**Authors:** Rasha S. Teleb, Amal Abdul-Hafez, Amira Othman, Ahmed El-Abd Ahmed, Abdelrahman A. Elsaid, Hattan Arif, Ahmed A. Zarea, Mohammed Abdulmageed, Hend Mohamed, Sherif Abdelfattah Ibrahim, Ranga P. Thiruvenkataramani, Tarek Mohamed, Masamitsu Kanada, Burra V. Madhukar, Myrna Gonzalez Arellano, Mohammed M. Sayed, Heba M. Qubaisy, Said A. Omar

**Affiliations:** 1Division of Neonatology, Department of Pediatrics and Human Development, College of Human Medicine, Michigan State University, East Lansing, MI 48824, USA; 2Department of Pediatrics and Neonatology, Faculty of Medicine, South Valley University, Qena 83523, Egypt; 3Regional Neonatal Intensive Care Unit, Sparrow Hospital, Lansing, MI 48912, USA; 4Clinical Pathology Department, Faculty of Medicine, South Valley University, Qena 83523, Egypt; 5The Institute for Quantitative Health Science & Engineering, College of Human Medicine, Michigan State University, East Lansing, MI 48824, USA; 6Cell and Molecular Biology Program, Michigan State University, East Lansing, MI 48824, USA; 7Histology and Cell Biology Department, Faculty of Medicine, Mansoura University, Mansoura 35516, Egypt; 8Department of Pharmacology and Toxicology, College of Human Medicine, Michigan State University, East Lansing, MI 48824, USA

**Keywords:** placenta, umbilical cord, umbilical cord blood transplantation, mesenchymal stem cells, hematopoietic stem cells, extracellular vesicles (EVs), exosomes

## Abstract

Background: Mesenchymal stem cells (MSCs) have been used for ex vivo expansion of umbilical cord blood (UCB) hematopoietic stem cells (HSCs) to maintain their primitive characters and long-term reconstitution abilities during transplantation. Therapeutic effects of MSCs mainly rely on paracrine mechanisms, including secretion of exosomes (Exos). The objective of this study was to examine the effect of cord blood plasma (CBP)-derived Exos (CBP Exos) and Placental MSCs-derived Exos (MSCs Exos) on the expansion of UCB HSCs to increase their numbers and keep their primitive characteristics. Methods: CD34^+^ cells were isolated from UCB, cultured for 10 days, and the expanded HSCs were sub-cultured in semisolid methylcellulose media for primitive colony forming units (CFUs) assay. MSCs were cultured from placental chorionic plates. Results: CBP Exos and MSCs Exos compared with the control group significantly increased the number of total nucleated cells (TNCs), invitro expansion of CD34^+^ cells, primitive subpopulations of CD34^+^38^+^ and CD34^+^38^−^Lin^−^ cells (*p* < 0.001). The expanded cells showed a significantly higher number of total CFUs in the Exos groups (*p* < 0.01). Conclusion: CBP- and placental-derived exosomes are associated with significant ex vivo expansion of UCB HSCs, while maintaining their primitive characters and may eliminate the need for transplantation of an additional unit of UCB.

## 1. Introduction

Umbilical cord blood transplantation (UCBT) has been used for the treatment of different hematological conditions with many advantages over bone marrow transplantation (BMT), such as fast availability of banked cryopreserved units, lower risk of infections, lack of risk to the donor, and lower incidence and severity of graft versus host disease (GVHD) [1]. Delayed or failed engraftment due to low cell dose is the main disadvantage of UCBT [2]. Multiple clinical studies have showed that the dose of 1–2.5 × 10^7^ total nucleated cells (TNCs)/kg and or 1–1.5 × 10^5^ CD34^+^ cell/kg are needed for better engraftment and survival, which is rarely achieved, particularly for patients with higher weight [3,4,5]. To overcome this problem, two units of UCB are usually needed for successful transplantation [6]. One major disadvantage of using multiple UCB units per patient is that it greatly decreases the availability of HLA-matched UCB, which is essential regardless of cell dose, to achieve good engraftment and overall survival [7,8]. Continuous efforts have been dedicated to increasing the number of HSCs in UCB units by in vitro expansion prior to infusion as an alternative to double UCB transplantation [7]. The goal of UCB hematopoietic stem cells (HSCs) expansion is to increase their number while maintaining their primitive characteristics in the expanded population [9]. These primitive cells, usually marked by their CD34^+^38^−^ surface markers, are important in long-term engraftment during UCB transplantation [10]. However, in vitro expansion in a liquid culture system usually favors the proliferation and differentiation of HSCs at the expense of their stemness. The loss of stemness in liquid HSCs cultures could be due to removal from their hematopoietic microenvironment, called stem cell niche, which contains important molecular cues that regulate different fates of HSCs: quiescence, self-renewal, proliferation, and differentiation [11,12,13,14]. Mesenchymal stem cells (MSCs) are stem cells that have been identified as one major component of the bone marrow (BM) hematopoietic microenvironment [12]. MSCs were found to secrete a broad range of molecules that regulate hematopoiesis. Using MSCs as a feeder layer for UCB HSCs in a co-culture system has shown to greatly enhance the proliferation of human HSCs, especially the more primitive CD34^+^38^−^ fraction [15,16,17,18]. MSCs co-administration with CB HSCs has also been shown to promote the engraftment of human CD34^+^ cells in immunodeficient (NOD/SCID) mice [19]. Clinical trials have also reported a reduction in the neutrophils’ and platelets’ recovery time after transplantation of two units of UCB, one of which is expanded in a co-culture with BM MSCs [20,21]. In addition, MSCs have been shown to have a clinical immunomodulatory activity that may impact GVHD in transplant patients [22,23]. Among the various sources of MSCs, those derived from gestational tissues (fetal stem cells) have the advantage of being a feasible unlimited source of stem cells with high plasticity, rapid proliferation, and low risk for transmission of infection [24]. Gestational tissues, such as the placenta and umbilical cord, are considered as medical waste, so their use in research raises no concerns about harming the donor or other ethical problems [24]. Several studies reported a co-culture of UCB HSCs with fetal MSCs from the placenta, Wharton Jelly (WJ), and UCB increased the number of HSCs with the primitive character and long-term reconstitution ability in animal models [25,26]. Accumulating evidence suggests that the therapeutic activity of MSCs is mainly attributable to their paracrine effects. MSCs secrete bioactive components such as growth factors and cytokines, either directly to the environment or as components of vesicles, broadly termed extracellular vesicles (EVs) [27,28,29,30,31]. EVs are heterogeneous populations of nano-sized membrane-enclosed fragments of cytoplasm and bioactive materials (mRNA, miRNA, DNA, protein, lipid, and other small molecules) that are produced by nearly all eukaryotic cells [32]. The main active component of EVs is thought to be small EVs or exosomes, which range in size from 50 to 150 nm in diameter and are derived from the endosomal compartment, with other microparticles such as ectosomes and apoptotic bodies [33]. Exosomes have an essential role in regulating BM functions during homeostasis [34]. Several studies suggested that exosomes derived from MSCs could support the expansion of HSCs with the same beneficial effects as those of MSCs co-culture [35,36,37]. Besides being a source of HSCs, UCB also contains other cells, such as immune cells and MSCs [38]. The immune-regulatory cells in UCB, such as regulatory T cells (Tregs) and myeloid-derived suppressor cells (MDSCs) [39,40], contribute to immune tolerance during pregnancy [41]. Exosomes released from UCB-derived cells into cord blood plasma (CBP) may have the same immune-regulatory effects as their parent cells [42,43]. CBP, if proved to be an equally effective source of exosomes as MSCs, has the advantage of being a readily procurable source from which exosomes could be extracted within hours from UCB collection and may be obtained from the same donor of HSCs. This is in contrast with MSCs, which need weeks to be cultured and processed for exosomes isolation. A limited number of studies have examined the structural and functional properties of CBP-derived exosomes [44,45,46]. Preclinical studies investigated their effect on cutaneous wound healing [47], experimental autoimmune encephalitis [48], and a mouse model of liver fibrosis [49] with encouraging results. To our best knowledge, CBP-derived exosomes have not been tested previously on the expansion of HSCs.

In this study, we investigated the effect of exosomes (50–150 nm in diameter) with specific surface markers (CD63, TSG1, Rab27a, and Flotillin-1) derived from both PL MSCs culture supernatant and CBP on the expansion of UCB HSCs. The main aim of this study was to determine the potential role of CBP-derived and PL MSCs-derived exosomes in enhancing the expansion of UCB HSCs in vitro without the loss of HSCs stemness. This will help to overcome the problem of low cell dose that limits the success of UCB transplantation and will eliminate the need for transplantation of an additional unit of UCB.

## 2. Methods

### 2.1. Sample Collection, Preparation, and Isolation of UCB CD34^+^ Cells

Figure 1 below shows the summary of the methods used in our study. After informed consent was obtained from the mothers, placentas and their UCB from healthy full term (FT) pregnancies were collected after delivery following protocols approved by the institutional review boards at Michigan State University, East Lansing and Sparrow Hospital, Lansing, MI, USA.

Mononuclear cells (MNCs) were isolated by density gradient centrifugation (Lymphoprep: STEMCELL Technologies, Vancouver, BC, Canada). After centrifugation, plasma samples were collected and immediately processed for exosomes isolation or stored in aliquots at −80 °C for later processing. MNCs were then aspirated and washed with PBS to remove excess Lymphoprep or plasma. CD34^+^ cells were isolated from MNCs using magnetic positive selection (EasySep™ Human Cord Blood CD34 Positive Selection Kit II: STEMCELL Technologies, Vancouver, BC, Canada) following the manufacturer’s instructions.

### 2.2. Placental Mesenchymal Stem Cells (MSCs) Culture

The placenta samples were processed to prepare for MSCs isolations on the same day of sample collection, and MSCs passages were completed for the next 3–5 weeks.

The placental chorionic plate tissue was used to isolate MSCs according to previously published protocols [50]. Briefly, fetal side of the placental chorionic plate tissue were minced into 1–2 mm pieces. Tissue pieces were treated with commercial trypsin solution (Trypsin 0.25% EDTA; Thermo Fisher Scientific, Waltham, MA, USA) at 37 °C for 30 min in a 5% CO_2_ incubator for partial digestion of the samples. The partially digested tissue pieces were plated in 75 cm^2^ culture flasks using a growth medium containing high-glucose Dulbecco’s modified Eagle’s medium (DMEM, Glutamax; Gibco, Carlsbad, CA, USA), supplemented with 10% fetal bovine serum (FBS; Thermo Fisher Scientific, Waltham, MA, USA), and 1% of antibiotic antimycotic solution; 10,000 units/mL of penicillin, 10,000 µg/mL of streptomycin, and 25 µg/mL of Amphotericin B (Thermo Fisher Scientific, Waltham, MA, USA) and incubated at 37 °C in an atmosphere of 5% CO_2_ in a humidified incubator. The outgrowth of cells from explants was observed by microscopy for characteristics of MSCs morphologies. Cells were passaged upon reaching 70–80% confluency and characterized using flow cytometry.

EV-depleted serum was prepared by 18 h ultracentrifugation at 100,000× *g* at 4 °C [51]. For each of passages 3–5, MSCs were seeded at a density of 1 × 10^4^ cells/cm^2^ in complete growth medium for 24 h and replaced by an EV-depleted medium to obtain the culture supernatant of MSCs (DMEM-high glucose supplemented with 10% EV-depleted serum and 1% antibiotic antimycotic). The conditioned medium was collected after 48 h for isolation of exosomes.

### 2.3. Isolation and Identification of Exosomes

Exosomes were isolated from CBP and the conditioned media (CM) of PL MSCs. Briefly, aliquoted samples were centrifuged at 600× *g* for 10 min to remove any cells and debris. The supernatant was centrifuged again at 2000× *g* for 30 min to remove apoptotic bodies. Supernatants were filtered through 0.22 µm membrane filters with pressure to remove large EVs. Exosomes were collected by size based EVs isolations method with modifications using 50 nm membrane filters (EMD Millipore, VMWP02500 or Whatman 110603) with holders (EMD Millipore, SX0002500) [52]. Briefly, the diluted CBP (diluted 1:3 in PBS) or the CM was added to a 10 mL syringe connected to the holder with a 50 nm membrane and filtered by applying vacuum, which was connected from the other side to a vacuum manifold and allowed to filter until only about 500 µL of the fluid remained in the holder. Then 5 mL of PBS was added to the syringe for washing and allowed to filter until only about 500 µL remained. Then the concentrated sample was collected from the holder and stored at −80 °C until use.

Size range, morphology, and protein markers of the collected exosomes were analyzed by nanoparticle tracking analysis (NTA), transmission electron microscopy (TEM), and western blotting (WB).

#### 2.3.1. Nanoparticle Tracking Analysis (NTA)

NTA was carried out using the Zeta View (Particle Metrix, Analytik Ltd., Cambridge, UK) following the manufacturer’s instructions. Exosomes derived from CBP and PL MSCs were further diluted 100- to 1000-fold with PBS for the measurement of particle size and concentration.

#### 2.3.2. Transmission Electron Microscopy (TEM)

Samples were prepared as previously reported [52,53]. Isolated exosomes were fixed in 2% paraformaldehyde for 5 min. For negative staining of exosomes, 5 mL of the sample solution was placed on a carbon-coated EM grid and exosomes were immobilized for 1 min. The grid was transferred to five drops of distilled water 100 µL each and letting it on the surface of each drop for 2 min sequentially. The sample was negatively stained with 1% uranyl acetate. The excess uranyl acetate was removed by contacting the grid edge with filter paper and the grid was air dried. The grids were imaged with a JEOL100CXII Transmission Electron Microscope operating at 100 kV. Images were captured on a Gatan Orius Digital Camera.

#### 2.3.3. Western Blotting

Equal total protein derived either from CBP or PL MSCs exosomes were mixed with 5× Pierce™ Lane Marker Reducing Sample Buffer (Thermo Fisher Scientific, Waltham, MA, USA). The following four primary antibodies were used for the common exosomes surface markers: CD63, TSG101, Rab27a, and Flotillin1. ECL detection was done using ECL Select Western Blotting Detection Reagent (GE Healthcare, RPN2235) on ChemiDoc Imaging System (Bio-Rad Laboratories Inc., Hercules, CA, USA).

### 2.4. Expansion of HSCs

Isolated CD34^+^ cells were cultured at 2 × 10^4^ cells/mL (24 wells plate, 1.1 mL/ well) for 10 days under various culture conditions. The culture conditions were classified into three groups: (1) The control group: Serum-free HSCs expansion medium (Stemline II Hematopoietic Stem Cell Expansion Medium; Sigma-Aldrich, St. Louis, MO, USA) supplemented with ready recombinant cytokine mix: thrombopoietin (TPO), stem cell factor (SCF), FMS-like tyrosine kinase (3flt-3) ligand and interleukin (IL)-3 (Hematopoietic Progenitor Expansion Medium DXF; Sigma-Aldrich), (2) The CBP exosomes group: Serum free HSCs expansion medium supplemented with recombinant cytokine mix (same as above) and CBP exosomes (final protein concentration = 100 µg/mL), and (3) The PL MSCs-derived exosomes group: Serum free HSCs expansion medium supplemented with recombinant cytokine mix (same as above) and PL MSCs-derived exosomes (final conc. = 100 µg/mL). The dose of exosomes was chosen by dose escalation of protein concentration of 5, 10, and 100 µg/mL which showed that the dosage of 100 µg/mL has the most significant effect in expanding the number of CD34^+^ cells in cell culture.

Fresh medium with or without cytokines were exchanged twice a week in all groups. On days 3, 7, and 10 of expansion, the cells were counted to determine the total cell number. Cells on day 0 and day 10 were examined for surface markers of primitive and differentiated HSCs by flow cytometry (Cytec Aurora, Cytec Biosciences, Fremont, CA, USA).

### 2.5. Immunophenotyping

Flow cytometric analyses were performed to: (1) characterize isolated placenta cells using surface markers defining MSCs; and (2) determine the phenotype of the pre and post expansion UCB CD34^+^ cells for the primitive cell markers (CD34^+^CD38^−^ and CD34^+^CD38^−^Lin^−^) and differentiated cells markers (CD45^+^Lin^+^).

For PL MSCs characterization, the harvested placental cells were stained with the following fluorescent antibodies: anti-CD44 (AF700), anti-CD73 (BV785), anti-CD90 (PE.CY7), anti-CD105 (FITC) (BioLegend, San Diego, CA, USA), and PE-labeled negative selection cocktail (anti-CD34, -CD45, -CD11b, -CD19, and -HLA-DR antibodies) (BD Biosciences, San Jose, CA, USA).

Cells then were analyzed by flow cytometer (Cytec Aurora, Cytec Biosciences). For the analysis of UCB CD34^+^ cells before and after expansion on day 0 and day 10, cells were stained with the following antibodies: anti-CD45 (BV 510); anti-CD38 (BV 650) (BioLegend, San Diego, CA, USA); anti-CD34 (PE); and FITC labeled cocktail for differentiation markers: CD2, CD3, CD4, CD7, CD8, CD10, CD11b, CD14, CD19, CD20, CD56, CD235a (Human Lineage Cocktail 4 [lin 4]), both from BD Bioscience (San Jose, CA, USA). Cells then were analyzed by flow cytometry (Cytec Aurora, Cytec Biosciences). The proportions of CD45^+^Lin^+^, CD34^+^CD38^−^, CD34^+^CD38^−^Lin^−^ cells are indicated as percentages of the total analyzed cells. The absolute numbers of these cell populations were calculated according to the mean numbers of the selected cells before and after expansion.

### 2.6. Colony Forming Unit (CFU) Assay

For the various culture conditions, duplicate assays were performed for Human Colony Forming Unit (CFU) Assays Using MethoCult™ according to the manufacturer’s protocol (StemCell Technologies, Vancouver, BC, Canada). Briefly, cells were harvested after 10 days of expansion and 10^4^ of the expanded cells were cultured in MethoCult GF H4435 methylcellulose medium for 14 days in an atmosphere of 5% CO_2_ and 95% air at 37 °C, following the manufacturer’s instructions. The numbers of colony forming unit-erythroid (CFU-E), burst forming unit- erythroid (BFU-E), colony forming unit-granulocytes/macrophages (CFU-GM), and colony forming unit granulocytes/erythrocytes/monocytes/megakaryocyte (CFU-GEMM) were counted under an inverted microscope (TCM400, Labo America Inc., Fremont, CA, USA).

## 3. Statistics

The results of the study were analyzed using SigmaPlot statistical software version 14.5 (Systat software Inc., San Jose, CA, USA) by the Student’s *t*-test or analysis of variance (ANOVA) to evaluate statistical significance between means. All pairwise multiple comparisons were done using the Student–Newman–Keul method. Results were statistically significant if *p*-value < 0.05. Results were reported as mean ± SD or mean ± SEM.

## 4. Results

### 4.1. Placental Samples

A total of 12 samples of placenta and their cord blood were obtained from healthy full-term pregnancies with a mean gestational age of 39 ± 1 weeks and no evidence of maternal hypertension, preeclampsia, diabetes mellitus, chorioamnionitis, or chronic conditions.

### 4.2. MSCs Culture and Identification

The cultured placental cells were found to exhibit mesenchymal morphological features: adherent and spindle-shaped fibroblast-like cells. PL MSCs were expanded to passage 5. An average of 0.4 × 10^6^ cells were obtained per gram of placental tissue. By flow cytometry, the isolated placenta cells (passage 3) displayed positive expression for MSCs markers (mean ± SD): CD73 (98 ± 2 %), CD 90 (97 ± 3%), CD105 (94 ± 5%), and CD44 (99 ± 5%). MSCs showed the following stem cells markers: OCT4 (12 ± 7%), SOX2 (50 ± 26%), and less than 1% of cultured cells showed expression of hematopoietic markers CD34, CD45, CD11b, CD19, and HLA-DR.

### 4.3. Identification and Analysis of Isolated Exosomes

EVs isolated from CBP and CM of PL MSCs were identified by NTA, TEM, and Western blot analysis of exosome marker proteins. NTA showed that the mean diameters of the isolated particles from both sources were around 100 nm. The average concentration in CBP Exos was 5 × 10^10^ EVs/mL (range 0.2–11 × 10^10^ Evs/mL), while it was 2 × 10^10^ EVs/mL (range 1.4–3.9 × 10^10^ EVs/mL) from one million seeded MSCs (passage 3–5) (Figure 2a). The results of the Western blot revealed the presence of exosomes specific markers (CD63, TSG 101. Rab27A, and flotillin) in these isolated nanoparticles (Figure 2b). TEM showed that both CBP and PL MSCs derived particles exhibited a cup- or round-shaped morphology with diameters ranging from 50 to 150 nm (Figure 2c). These data are consistent with the previously reported characteristics of exosomes.

### 4.4. The Effects of CBP and PL MSCs Exosomes on Ex Vivo Expansion of CD34+ to Total Nucleated Cells (TNCs)

The Mean ± SD number of total nucleated cells (TNCs) after 10 days in culture was significantly higher in the CBP- and PL MSCs-derived exosomes compared with the untreated control group (23 ± 6 and 23 ± 5 vs. 15 ± 2 × 10^5^; *p* < 0.0001) as seen in Figure 3.

### 4.5. Flow Cytometry of the Expanded HSCs

Flow cytometric analysis was performed to evaluate the effects of CBP and Pl MSCs exosomes on the surface marker profile of the expanded CD34+ cells on day 10 to assess their stemness as seen in Figure 4.

### 4.6. The Effects of CBP and PL MSCs Exosomes on Ex Vivo Expansion of CB CD34^+^ Cells

The average amount of UCB collected for the study was 62 mL (range of 25–125 mL) and the average number of CD34^+^ cells isolated from the study UCB samples was 3.6 × 10^6^ (range 0.2−4.4 × 10^6^). The CD34^+^ cells were divided into three groups: untreated CD34^+^ cells as control, CD34^+^ with CBP-derived exosomes, and CD34^+^ with PL MSCs-derived exosomes as described above. The results showed that the absolute number of CD34^+^ cells after 10 days of expansion in culture was significantly higher in the PL MSCs and CBP Exos compared with the untreated control group (20-fold and 14-fold vs. 11-fold; *p* < 0.001 over the number of CD34^+^ cells (2 × 10^4^) seeded prior to culture) as seen in Figure 5a.

### 4.7. The Effects of CBP and PL MSCs Exosomes on Ex Vivo Expansion of Primitive HCSs Cells

There was a statistically significant increase in the absolute number of the most primitive sub-fraction CD34^+^38^−^ cells after 10 days in culture in the CBP and PL MSCs exosomes compared with the untreated control group (80-fold and 85-fold vs. 43-fold, *p* < 0.0001), as seen in Figure 5b. There was also a statistically significant increase in the absolute number of the CD34^+^38^−^Lin^−^ cells after 10 days in culture in the CBP and PL MSCs exosomes compared with the untreated control group (37-fold and 47-fold vs. 28-fold, *p* < 0.0001), as seen in Figure 5c. This indicates that exosomes from both sources were able to keep the stemness of the expanded cells (Figure 5b,c). In addition, there was also a statistically significant increase in the absolute number of the expanded differentiated CD45^+^Lin^+^ cells after 10 days in culture in the CBP and PL MSCs exosomes compared with the untreated control group (380-fold and 396-fold vs. 260-fold, *p* < 0.0001), as seen in Figure 5d.

### 4.8. The Effects of CBP and PL MSCs Exosomes on the Generation of CFUs from the Expanded CD34^+^ Cells

The CFUs assay was performed to measure the frequency of progenitor cells that were able to develop into colonies of blood cell lines. After 10 days of expansion, cells under various culture conditions were harvested and seeded in semisolid methylcellulose for another 14 days, as described above. Morphological analysis showed that the total number of CFUs was significantly higher in the CBP-derived exosomes and PL MSCs-derived exosomes compared with the untreated control group after 14 days (24 ± 2 and 23 ± 2 vs. 16 ± 2 CFUs; *p* < 0.01). In addition, the expanded cells in all groups were able to produce the different types of hematological colonies: CFU-E, BFU-E, CFU-G, CFU-M, CFU-GM, and CFU-GEMM as seen in Figure 6a,b.

## 5. Discussion

The low cell numbers present in UCB has limited its use as a source for HSCs in transplantation procedures. It is estimated that a minimum dose of 0.7–1.5 × 10^5^ CD34^+^ cells/kg of patient body weight is required for successful single unit transplantation [56], which is rarely achieved in UCB units except for young children.

Double CBT is limited by the ability to find HLA matched units and significantly increased engraftment failure [57]. Increasing the cell dose in the CB unit before transplantation was found to greatly improve neutrophil and platelet counts and reduce engraftment failure [57]. This was the rationale behind the need to optimize the conditions during ex vivo expansion of CB HSCs to obtain the best results. In addition, the key to life-long HSCs maintenance is to make sure that the balance between differentiation and self-renewal is in favor of the latter as every cell division represents a potential threat for the more primitive HSCs to be depleted [58]. If the ex vivo expansion provides a sufficient number of cells that are also capable of long-term support of the hematopoiesis (long-term repopulating cells), then the expanded cell population will contain both undifferentiated and committed cells, which can guarantee short-term and long-term recovery of hematopoiesis after the transplantation.

The role of BM MSCs as a component of HSCs niche in the regulation of different aspects of hematopoiesis is now well recognized [11]. Furthermore, the use of MSCs as a feeder layer during UCB HSCs expansion proved to enhance the number and stemness of expanded cells, which could enhance the results of UCB transplantation [57]. BM MSCs may be the cells of choice for stromal support of HSCs as they act as a natural scaffold for the hematopoietic cells in their original niche. However, their use is limited by the pain and risk of infection associated with their harvest.

MSCs from fetal placenta and UCB may act as a successful alternative that is reported to have the same effect as BM MSCs without the risk of harm to the donor and a reported higher rate of proliferation and lower immunogenicity [17,25,26].

One main limitation for using MSCs (from BM of the patient or related donor or from gestational tissues) as a feeder layer for HSCs expansion is the long time needed for culturing MSCs to produce enough number of cells (about 3–5 weeks) in addition to the time needed for the HSCs expansion after that [57]. This is a major obstacle for the use of this technique in the treatment of many hematological diseases such as in hematological malignancies in which time is critical.

It has been demonstrated that MSCs can secrete or express a broad range of molecules that can regulate various aspects of hematopoiesis [11]. It was also found that, apart from soluble factors, MSCs can also secrete a large number of exosomes that act as important mediators of cell-to-cell communication [28,32,35].

Cell-free therapy using exosomes is a rapidly increasing field in regenerative medicine. Exosome therapy offers several advantages over the use of their parental cells such as lack of risk of aneuploidy or immune reactions, and the surface or contents could be engineered to enhance disease-specific targeting [59]. All these factors could favor the use of exosomes as ready “off-the-shelf” therapeutics in clinical practice in the future.

Assuming that exosomes released from different stem cells in UCB and placenta may mimic the hematopoiesis-supporting effects of their parental cells, we investigated the effect of those exosomes on the expansion of HSCs regarding increasing the number of cells and in the meantime keeping their primitive characters, which is essential for successful long-term engraftment after CBT.

To the best of our knowledge, this is the first time that the possibility of using exosomes from CBP and PL MSCs to improve the ex vivo expansion of UCB HSCs has been studied. The current study established two major findings.

First, our protocol for the isolation and culture of placenta MSCs was found to be reproducible at each time and resulted in the production of a large number of MSCs with typical morphology and surface marker profile. Similar results were reported using the same explant culture method to isolate MSCs from gestational tissues from different research groups [50,60,61]. The MSCs numbers obtained from the whole placenta, without serial culture, are much higher than the MSCs derived from bone marrow; additionally, the other limitations such as painful and invasive procedure and the scarcity and difficulty in expanding the BM MSCs population can be avoided [62].

Second, abundant exosomes were isolated from CBP in the same day of HSCs isolation from the CB, while it took 3–5 weeks to obtain the exosomes from PL MSCs cultures. From the view of clinical application, CBP exosomes could be isolated the same day of CB unit collection and from the same donor. In blood banks, plasma could be easily stored together with their UCB cells. When the HSCs are needed, the plasma could be thawed, and exosomes could be ready within 6–8 h to be used for the expansion of HSCs if their numbers are low [63].

Our results show that CBP- and PL MSCs-derived exosomes have promoted the expansion of UCB CD34^+^ cells. Exosomes-treated groups generated significantly greater numbers of CD34^+^ cells compared with that in the untreated control group. The average yield of the CD34^+^ cells in the study’s CB samples is 3.6 × 10^6^ cells, using the current protocol of expansion could provide as high as 50.4 × 10^6^ (14 fold increase) in CD34^+^ cells expanded with CBP Exos and 72 × 10^6^ (20 fold increase) in CD34^+^ cells expanded with PL MSCs Exos, which offer more than 1.5 × 10^5^ of total CD34+ cell/kg previously demonstrated to be adequate for better engraftment, successful results in transplantation and higher survival for patients with weight higher than 45 kg [5,6].

To test the stemness of the expanded cells, they were analyzed by flow cytometry for the surface markers of primitive and differentiated cells. It is reported that the primary HSCs marked by the surface marker CD34^+^CD38^−^Lin^−^ are responsible for the successful long-term engraftment and can give rise to multilineage colonies containing both lymphoid and myeloid cells in SCID mice [64]. Our study showed that CBP- and PL MSCs-derived exosomes had higher number of the more primitive CD34^+^CD38^−^Lin^−^ cells than the untreated control group. The expanded cells contain larger absolute numbers of both primitive and differentiated cells which could be the basis of successful HSCs transplantation using one unit of UCB.

CFU assay was used to assess the ability of the expanded cells to form different hematological colonies in secondary cultures. CFUs of different types was found to be significantly higher in CBP exosomes and PL MSCs exosomes groups than the untreated control group (Figure 6b). This reflects the ability of the cells expanded in the exosomes groups to develop various hematological cell types.

Our results are in agreement with other studies done in this field which have shown that EVs derived from embryonic stem cells significantly improved the ex vivo expansion of murine HSCs with upregulation of the expression of the primitive HSCs markers [35]. Osteoblast-derived EVs have also been found to enhance the proliferation of UCB-derived CD34^+^ cells and retain their primitive functional cells in vitro and in vivo [36].

Xie H. and colleagues have reported that BM MSCs derived microvesicles (MVs) enhanced the ex vivo expansion of cord blood CD34^+^ HSCs and cord blood mononuclear cells with comparable results to MSCs-HSCs coculture system [37]. In addition, they reported that genomic analyses of adult BM MSCs-MVs revealed multiple miRNAs that are involved in the regulation of Wnt/β-catenin signaling pathway which is crucial for the regulation of hematopoiesis, promoting self-renewal and inhibiting HSCs differentiation [37].

Furthermore, EVs were found to have an important role in HSCs differentiation. HSCs exposed to MVs derived from megakaryocytes showed a dramatic induction of differentiation to mature megakaryocytes without exogenous addition of thrombopoietin [65]. Sarvar et al., also found that MSCs-derived MVs were able to reduce the erythroid differentiation of CD34^+^ cells derived from the UCB suggesting an important role for MSCs-derived EVs in the control of normal erythropoiesis [66].

Angulski et al., examined the proteomic analyses of human BM MSC-EVs and the expanded UCB CD133^+^ EVs to better understand the functions performed by these vesicles and cells and to delineate the most appropriate use of each type of EVs for future therapeutic procedures. Their data demonstrated that expanded CD133^+^ EVs and BM MSCs-EVs are in part similar but also sufficiently different to reflect the main beneficial paracrine effects widely reported in pre-clinical studies using expanded CD133^+^ cells and/or BM-MSCs [67].

The results of those studies along with ours, suggest that EVs may be one of the cues provided in vivo by MSCs that help maintain the functional properties of HSCs. However, there are many issues that need to be addressed in future work related to the use of exosomes for the ex vivo expansion of HSCs. For example, in the current study, we only performed an in vitro CFU assay to evaluate the potential of the expanded cells to develop into different blood cells, while the stemness of the cells can only be evaluated by in vivo transplantation assays [68]. Furthermore, the exact mechanisms by which exosomes exert their hematopoiesis-supporting effects are still not identified. Our future work will focus on studying the bioactive cargo of exosomes to identify which of the RNA or protein components could have regulatory effects on HSCs functions. In addition, future work is needed to examine the immunoregulatory function and immunosuppressive effects of placental MSCs vs. their derived EVs including exosomes in the clinical application for prevention or treatment of GVHD, which is a common fatal complication of allogeneic hematopoietic stem cell transplantation [69,70,71,72]. Finally, work should be done for optimizing the process of exosomes production on a larger scale and at a higher quality for clinical use [73].

## 6. Conclusions

In this study, we demonstrated that CBP and the conditioned media of PL MSCs could be valuable sources of exosomes and that exosomes facilitate ex vivo expansion of UCB HSCs, which may improve the results and eliminate the need for the use of additional UCB units during HSCs transplantation procedures.

The abundancy, ease of availability of both the CBP and placenta as sources of exosomes, and ultimately the timing from the start of sample collection to harvesting the expanded HSCs, is a point that may be crucial in case of treatment of some of the rapidly progressive hematological diseases such as acute leukemias.

## Figures and Tables

**Figure 1 cells-12-00250-f001:**
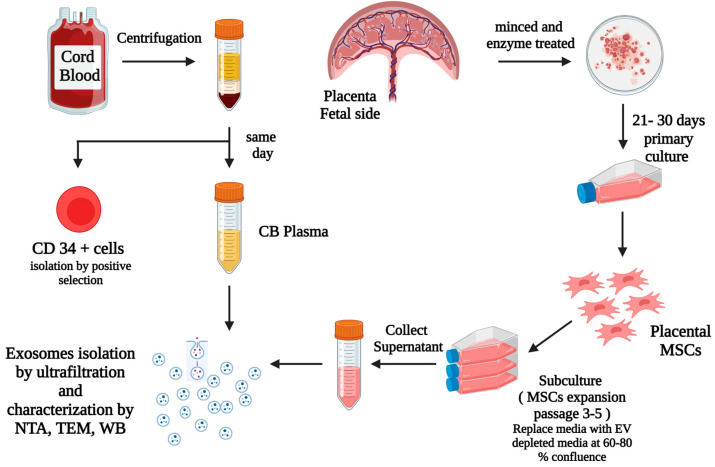
This diagram summarizes our methods and the steps for the isolation of CD34^+^ cells via positive selection from UCB, isolation, processing, and culture of placental tissues from the fetal side, isolation of primary MSCs (passage 0), expansion of MSCs via subculture from primary MSCs to passage 3–5, replacement of the culture media with EV depleted media when the expanded cells were at 60–80% confluence, then collection of the MSCs culture supernatant after 48 h and exosomes isolation from CBP and MSCs culture supernatants. Created with BioRender.com.

**Figure 2 cells-12-00250-f002:**
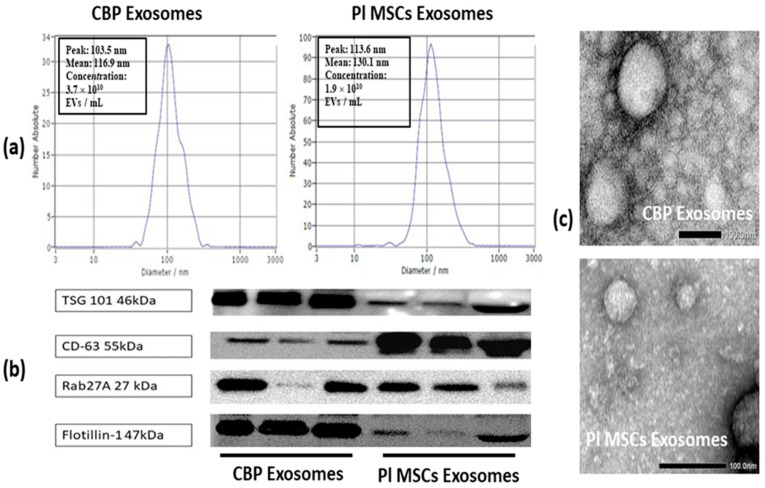
Characterization of exosomes derived from CBP and PL MSCs culture supernatant in our study. (**a**) Nanoparticles Tracking Analysis (NTA) was used to analyze the size and concentration of isolated exosomes. (**b**) Western blot analysis showing the presence of four common exosomes surface markers (CD63, TSG101, Rab27a, and Flotillin1) in both CBP and PL MSCs exosomes. (**c**) Typical morphology of CBP and PL MSCs exosomes observed under a transmission electron microscope (TEM). Scale bar = 0.2 µm. CBP: Cord blood plasma; PL MSCs: Placenta mesenchymal stem cells. The presence of exosomes is confirmed by the detection of at least four positive protein markers of EVs, including one transmembrane marker (CD63), lipid-bound protein (Flotillin-1), one cytosolic protein (TSG 101) [54], and a small cytosolic protein GTPase marker playing a key role in controlling intracellular transport and secretion of exosomes (Rab 27a) [55].

**Figure 3 cells-12-00250-f003:**
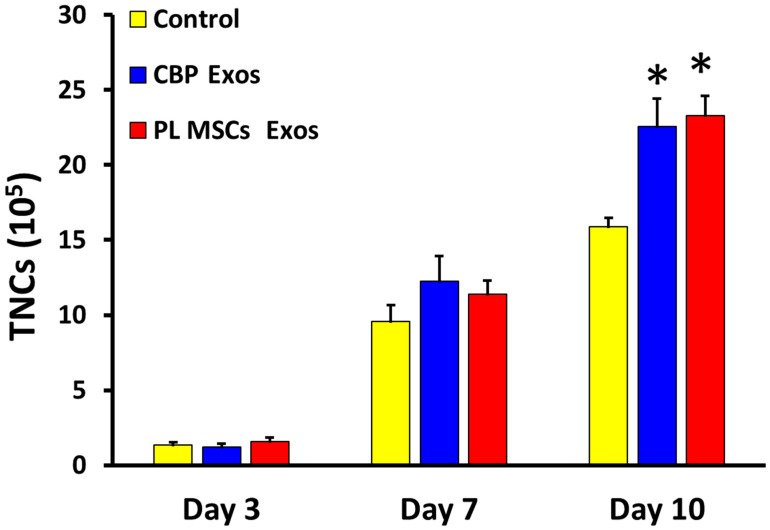
The effects of CBP and PL MSCs exosomes in the ex vivo expansion of TNCs after 10 days in culture. A total of 2 × 10^4^ CB CD34+ cells were cultured (pre-expansion) and CB CD34+ cells were maintained under three different culture conditions: untreated control, CBP Exos, and PL MSCs Exos (*n* = 4, 3 replicates each). TNCs were counted on days 3, 7, and 10 using the trypan blue exclusion method, compared with the untreated control group (* *p* < 0.001). CBP: cord blood plasma; PL MSCs: placenta mesenchymal stem cells.

**Figure 4 cells-12-00250-f004:**
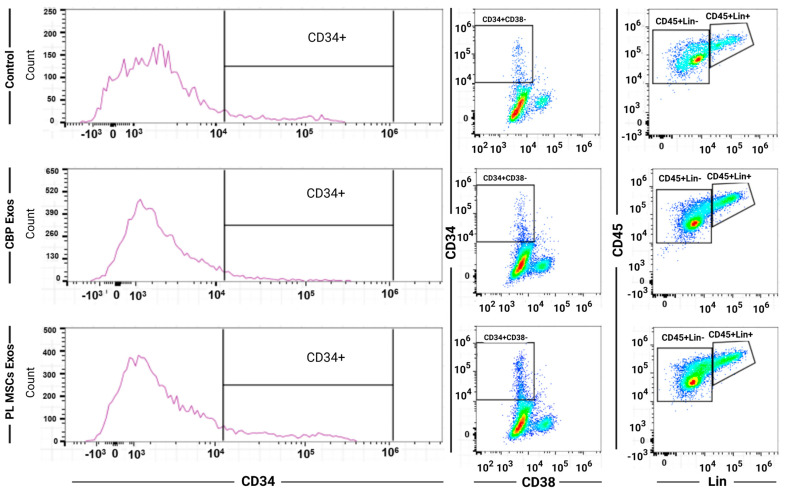
Representative flow cytometric dot blot diagrams of each subpopulation. CBP: cord blood plasma; PL MSCs: placenta mesenchymal stem cells. The effects of CBP and PL MSCs exosomes on the surface markers profile of the expanded CD34^+^ cells. CB CD34^+^ cells were maintained under three different culture conditions for 10 days (*n* = 3, 3 replicates each). Flow cytometric analysis for surface markers CD34, CD38, CD45, and Lin (surface markers cocktail of differentiated cells) was done for day 10 expanded cells.

**Figure 5 cells-12-00250-f005:**
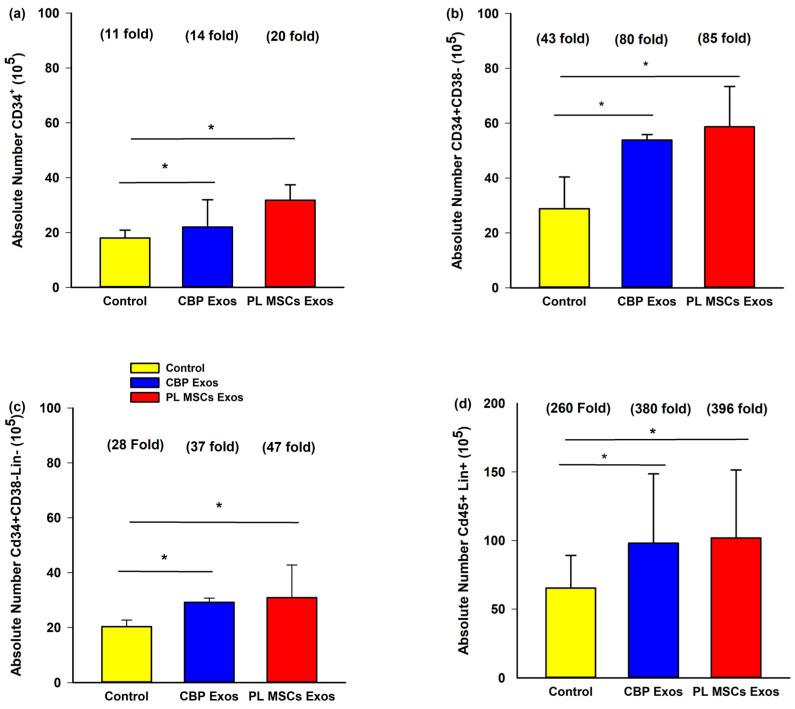
The effects of CBP and PL MSCs exosomes on the surface markers profile of the expanded CD34+. A total of 2 × 10^4^ CB CD34^+^ cells were cultured (pre-expansion) and maintained under three different culture conditions for 10 days (*n* = 3, 3 replicates each). Flow cytometric analysis for surface markers CD34, CD38, CD45, and Lin (surface markers cocktail of differentiated cells) was done for day 10 expanded cells: (**a**) Representative diagram for immunophenotype of subpopulations of expanded CB CD34^+^ cells; (**b**) representative diagram for immunophenotype of subpopulations of expanded CB CD34^+^ CD38^−^ cells; (**c**) representative diagram for immunophenotype of subpopulations of expanded CB CD34^+^ CD38^−^ Lin^−^ cells; (**d**) representative diagram for immunophenotype of subpopulations of expanded CB CD45^+^ Lin^+^ cells. ** p* < 0.001 compared with the control group. The fold increase represents the increase over the pre-expansion cells that were cultured.

**Figure 6 cells-12-00250-f006:**
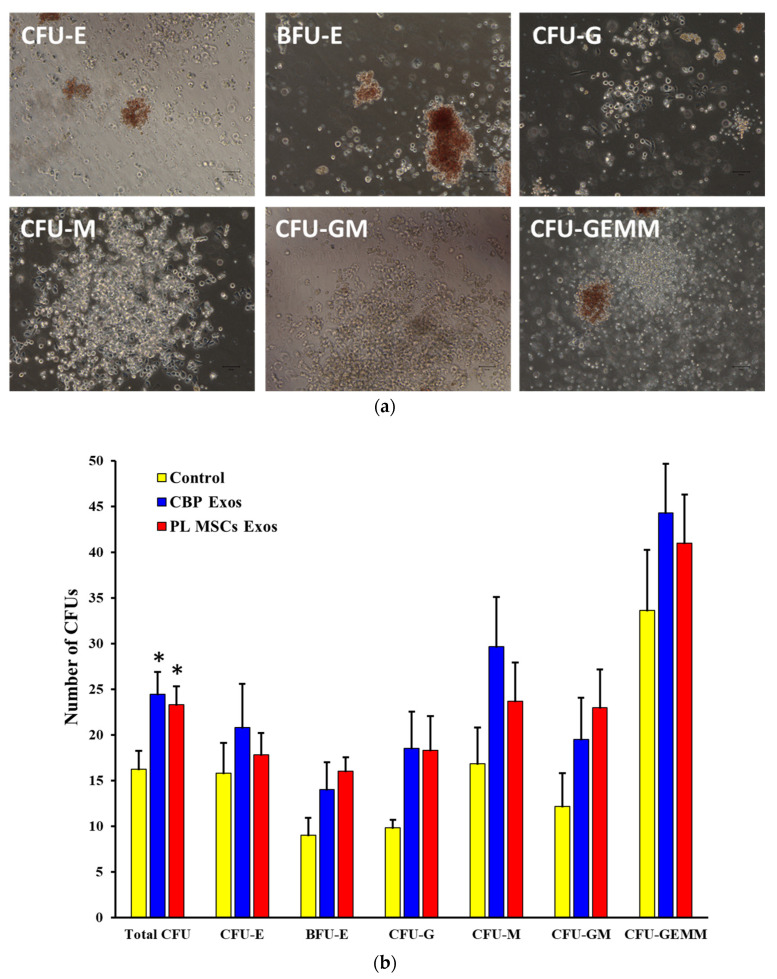
The effects of CBP and PL MSCs exosomes on the generation of CFUs from the expanded CD34^+^ cells. Day 10 expanded cells were plated in semi-solid cellulose media for another 14 days (*n* = 3, 3 replicates each). (**a**) Typical morphology of CFUs, observed using an inverted microscope. (**b**) Comparison of individual CFUs and average of total number of CFUs generated in the three groups after 14 days of expansion. CBP: cord blood plasma; PL MSCs-Exos: placenta mesenchymal stem cells-derived exosomes; CBP-Exos: cord blood plasma-derived exosomes. ** p* < 0.001.

## Data Availability

The authors have full control of all primary data, and the authors agree to allow the journal to review their data if requested. All data and material used for writing the manuscript are available.

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
