# Peer review of "Cord Blood Plasma and Placental Mesenchymal Stem Cells-Derived Exosomes Increase Ex Vivo Expansion of Human Cord Blood Hematopoietic Stem Cells While Maintaining Their Stemness"

_cells, 2023, doi:10.3390/cells12020250_

Round 1

Reviewer 1 Report

Overall Comments:

Umbilical cord blood transplantation (UCBT) has many advantages in the treatment of different hematological conditions over bone marrow transplantation (BMT) but low cell dose is the main problems in application. Based on accumulated reports saying that mesenchymal stem cells (MSCs) enhance proliferation of human UCB hematopoietic stem cells (HSCs), author in this manuscript treated the UCB-HSCs with extracellular vesicles (EV) isolated from fetal placenta derive MSCs or cord blood plasma (UCB) and examine the capability of increasing the proliferation. The EV derived from both sources enhanced proliferation of UCB-HSCs with retaining stemness as well as differentiation potential. Upon their results, authors suggest that the EV mediated increase in absolute cell numbers may increase the success rate in UCBT. Overall, author presented the results step by step with clear methods to support their claims. In addition, the issues they addressed are practically very important in clinical applications. Although several issues are revised, this manuscript is to be considered for publication in Cells if they answer all issues below.

Issues To Be Revised:

1.      Authors mentioned that the main aim of this manuscript was the influence of CBP derived EVs on proliferation and stemness of UCB HSCs. I think that the title may not stress that point. I recommend that author may emphasize the role of CBP EVs in title.

2.      On page 11 (line 231-234), authors said that the isolated placenta cells displayed positive expression for MSCs markers similar percentages to them expressed by standard BM-MSCs and less than 1% of cultured cells showed expression HSC markers. But they did not present any results or mention “data not shown”. Author may need present the results supporting their mention above. In addition, if they conducted any experiments comparing CBP-EVs and PL MSCs-EVs to BM-MSC/BM-MSC derived EVs, authors may need to show the data to support advantages of CBP-EVs.

3.      On page 11 (line 238-239), the values on Figure 2a were different from them in line 238. I wonder how much the figures are representative.

4.      On page 14, Y axis of the flow cytometric diagram for CD34 may need to be consistent. The different values of Y axis make the readers confused.

5.      On page 22 (line 397-399), author said that average yield of the CD34+ cells in one CB unit is 44x105 cells and their expansion protocol could provide 4.97-5.13x108 TNCs from one CB unit, which may offer higher chance for successful results in transplantation patients with higher weight. But if 1.0x 107 cells/kg is a dose threshold, the number of cells after expansion is, I think, still insufficient for patients over 50kg. You may need explain the sentence clearer.

6.      On page 22-23 (line 412-414), authors mentioned that the highest ability to form CFU of different types was found to be in CBP exosomes followed by PL MSCs exosomes. But based on Figure 4b, there was no significant difference between CBP and PL MSCs, which cannot support their claim.

7.      The resolution of each figure should be improved.

Typos

1.      On page 5 (line 116), please correct “Summarize” to “summarize”

2.      On page 16 (line 310), please correct “morphological” to “Morphological”

3.      On page 21 (line 387), please correct “Plasma” to “plasma”.

4.      Please correct “Pl MSCs” to “PL MSCs in each figure.

Author Response

Authors Response to Reviewer 1 Comments

Below are the authors’ response to the reviewers/s comments point by point in bold and as reflected in the submitted revised manuscript with track changes.

Umbilical cord blood transplantation (UCBT) has many advantages in the treatment of different hematological conditions over bone marrow transplantation (BMT) but low cell dose is the main problems in application. Based on accumulated reports saying that mesenchymal stem cells (MSCs) enhance proliferation of human UCB hematopoietic stem cells (HSCs), author in this manuscript treated the UCB-HSCs with extracellular vesicles (EV) isolated from fetal placenta derive MSCs or cord blood plasma (UCB) and examine the capability of increasing the proliferation. The EV derived from both sources enhanced proliferation of UCB-HSCs with retaining stemness as well as differentiation potential. Upon their results, authors suggest that the EV mediated increase in absolute cell numbers may increase the success rate in UCBT. Overall, author presented the results step by step with clear methods to support their claims. In addition, the issues they addressed are practically very important in clinical applications. Although several issues are revised, this manuscript is to be considered for publication in Cells if they answer all issues below.

Issues To Be Revised:

  1. Authors mentioned that the main aim of this manuscript was the influence of CBP derived EVs on proliferation and stemness of UCB HSCs. I think that the title may not stress that point. I recommend that author may emphasize the role of CBP EVs in title.

Response: The authors appreciate and agree with the reviewer comment.

The title was revised. The new title is “Cord Blood Plasma and Placental Mesenchymal Stem Cells -derived Exosomes Increase Ex Vivo Expansion of Human Cord Blood Hematopoietic Stem Cells While Maintaining Their Stemness

  1. On page 11 (line 231-234), authors said that the isolated placenta cells displayed positive expression for MSCs markers similar percentages to them expressed by standard BM-MSCs and less than 1% of cultured cells showed expression HSC markers. But they did not present any results or mention “data not shown”. Author may need present the results supporting their mention above. In addition, if they conducted any experiments comparing CBP-EVs and PL MSCs-EVs to BM-MSC/BM-MSC derived EVs, authors may need to show the data to support advantages of CBP-EVs.

Response: The authors appreciate and agree with the reviewer comment.

The authors added the percentage and SD of the identified MSCs markers at page 13, line 272-275) in the revised manuscript.

The authors did not design the study or conduct any experiments to compare the effect of CBP-EVs and PL MSCs-EVs to BM-MSC/BM-MSC derived EVs. This issue may be addressed in future studies.  

  1. On page 11 (line 238-239), the values on Figure 2a were different from them in line 238. I wonder how much the figures are representative.

Response: The authors appreciate the reviewer comment.

The graph represents one NTA sample from CBP exosomes and one sample from Pl MSCs exosomes to show the NTA sample graphs. The data in line 238 represents the average. To clarify further, in addition to the average, ranges of CBP and placental exosomes are added in the revised manuscript page 14 (line 279-280).

  1. On page 14, Y axis of the flow cytometric diagram for CD34 may need to be consistent. The different values of Y axis make the readers confused.

Response: The authors appreciate with the reviewer comment.

The Y axis range is different because with the CD 34+ cells expansion the number (count) of CD34+ was much higher in the CBP Exos and MSCs Exos after than the control group. This flow cytometry Y axis could not be changed, but the graph was replaced with a better resolution graph.

  1. On page 22 (line 397-399), author said that average yield of the CD34+ cells in one CB unit is 44x105cells and their expansion protocol could provide 4.97-5.13x108 TNCs from one CB unit, which may offer higher chance for successful results in transplantation patients with higher weight. But if 1.0x 107 cells/kg is a dose threshold, the number of cells after expansion is, I think, still insufficient for patients over 50kg. You may need explain the sentence clearer.

Response: The authors appreciate  the reviewer’s comment.

This section was revised for more clarification, to make it clearer and the following was added in page 26-27 in the revised manuscript (line 455-460); The average yield of the CD34+ cells per CB unit in our samples is 3.6 × 106 cells, using the current protocol of expansion could provide as high as 50.4 X 106 (14 fold increase) in CD34+ cells expanded with CBP-Exos and 72 X 106 (20 fold increase) in CD34+ cells expanded with Pl MSCs-Exos, which offer more than 1.5 x 105 of total CD34+ cell/kg previously demonstrated to be adequate for better engraftment, successful results in transplantation and higher survival for patients with weight higher than 45 kg. 

  1. On page 22-23 (line 412-414), authors mentioned that the highest ability to form CFU of different types was found to be in CBP exosomes followed by PL MSCs exosomes. But based on Figure 4b, there was no significant difference between CBP and PL MSCs, which cannot support their claim.

The authors agree with the reviewer and this section was revised.

The following as added to the revised manuscript in page 29 (line 470-472):

CFU of different types was found to be significantly higher in CBP exosomes and PL MSCs exosomes groups  than the untreated control group (Figure 6b). This reflects the ability of the cells expanded in exosomes containing conditions to develop various hematological cell types.

  1. The resolution of each figure should be improved.

 As suggested by the reviewer, the resolution of all figures was improved.

The following Typos were corrected

  1. On page 5 (line 116), please correct “Summarize” to “summarize”

This typo was corrected.

  1. On page 16 (line 310), please correct “morphological” to “Morphological”

This typo was corrected.

  1. On page 21 (line 387), please correct “Plasma” to “plasma”.

This typo was corrected.

  1. Please correct “Pl MSCs” to “PL MSCs in each figure.

This typo was corrected in all figures.

Reviewer 2 Report

In the manuscript of R.S. Teleb et al. “Placental Exosomes Increase Ex Vivo Expansion of Human Cord Blood Hematopoietic Stem Cells While Maintaining Their Stemness” authors developed the approach for improving the characteristics of umbilical cord blood hematopoietic stem cells (UCB HSCs) which are used in medicine for curative transplantation in a number of malignant and non-malignant diseases. To get over shortcomings related to delayed engraftment of HSCs and their low proliferation velocity, with simultaneous saving their primitive state, authors performed HSC treatment by extracellular vesicles (exosomes)  isolated from cord blood plasma and conditioned media of placental mesenchymal stem cells.

The data characterizing the quality of exosomes, as well as the exosome-induced enhancement of HSCs proliferation with preservation of their stemness and colony-forming capacity, are convincing and allow us to qualify this work as relevant.

At the same time, this paper cannot be published at its present form because of multiple inaccuracies.

Reviewer’s remarks.

Fig. 1 is poorly understandable. Within the picture, all samples should be marked as “USB”, “Culture of placental cells” “MSC expansion” and so on, otherwise it could be misunderstood. The legend should be also more detailed.

Fig. 4  Clarity of the picture (sharpness should be enhanced.

Fig. 5. What is “pre-treatment”? Why it is absent in fig. 3?!

Fig. 6. The design style of this figure stands out from the general style of the manuscript. In this relation, authors are recommended:

-          to mark pictures simply as “a” and “b”;

-          to insert “CFU-E, BFU-E, CFU-G” etc. instead of letters

-          to make a link to the transcription of the designations in the Methods (“Colony forming assay”)

-          In Methods (“Colony forming assay”) represent a complete list of abbreviations (to add “CFU-G”, CFU-G)…

Minor remarks: Some phrases should be edited.

Line 33

…Objective: was to examine the effect of cord blood plasma (CBP)…

Line 129

…Briefly, fetal side placental chorionic plate tissue pieces were minced into approximately 1–2 mm pieces …..

Line 132

…tissue pieces were plated in 75 cm2 culture flasks for explant culture using a growth medium containing high-glucose Dulbecco’s modified Eagle’s medium (DMEM, Glutamax;…

Line 260

as seen in figure 2 = FIGURE 3

line 438

…in the clinical application for prevention or treatment Graft-versus-host disease (GVHD) which…-authors already used this abbreviation in the text.

Line 477

Sparrow Hospital Clinical Research Institute

Authors are strongly recommended to check other misprints in the text of manuscript.

Author Response

Reviewer 2 Comments

Comments and Suggestions for Authors

In the manuscript of R.S. Teleb et al. “Placental Exosomes Increase Ex Vivo Expansion of Human Cord Blood Hematopoietic Stem Cells While Maintaining Their Stemness” authors developed the approach for improving the characteristics of umbilical cord blood hematopoietic stem cells (UCB HSCs) which are used in medicine for curative transplantation in a number of malignant and non-malignant diseases. To get over shortcomings related to delayed engraftment of HSCs and their low proliferation velocity, with simultaneous saving their primitive state, authors performed HSC treatment by extracellular vesicles (exosomes)  isolated from cord blood plasma and conditioned media of placental mesenchymal stem cells.

The data characterizing the quality of exosomes, as well as the exosome-induced enhancement of HSCs proliferation with preservation of their stemness and colony-forming capacity, are convincing and allow us to qualify this work as relevant.

At the same time, this paper cannot be published at its present form because of multiple inaccuracies.

Reviewer’s remarks.

Fig. 1 is poorly understandable. Within the picture, all samples should be marked as “USB”, “Culture of placental cells” “MSC expansion” and so on, otherwise it could be misunderstood. The legend should be also more detailed.

Response: The authors appreciate the reviewer’s comment.

The figure was revised with more labelling and easier follow through. As suggested by the reviewer. more details were added to the figure legend with clear explanation for better understanding.

Fig. 4  Clarity of the picture (sharpness should be enhanced.

As suggested by the reviewer, the resolution of this figures and all other figures were enhanced.

Fig. 5. What is “pre-treatment”? Why it is absent in fig. 3?!

The authors appreciate the reviewer comment.

Pre-treatment refers to the pre-expansion cell number. To avoid confusion, this part (pretreatment) of the graph was removed, and the pre-expansion cell number plated in cell culture was added to the figure legend.

The pretreatment is the preexposure or the initial number of CD34 cells (2 x 104) which were seeded in the cell culture. This was added to the figure legend.

Fig. 6. The design style of this figure stands out from the general style of the manuscript. In this relation, authors are recommended:

-          to mark pictures simply as “a” and “b”;

As suggested by the reviewer the pictures were marked as a and b.

-          to insert “CFU-E, BFU-E, CFU-G” etc. instead of letters

As suggested by the reviewer, colony forming units “CFU-E, BFU-E, CFU-G” etc. were inserted in the picture instead of letters.

-          to make a link to the transcription of the designations in the Methods (“Colony forming assay”)

This was added as suggested.

-          In Methods (“Colony forming assay”) represent a complete list of abbreviations (to add “CFU-G”, CFU-G)…

This was added as suggested.

Minor remarks: Some phrases should be edited.

Line 33

…Objective: was to examine the effect of cord blood plasma (CBP)…

As suggested by the reviewer this was edited.

Line 129

…Briefly, fetal side placental chorionic plate tissue pieces were minced into approximately 1–2 mm pieces …..

As suggested by the reviewer this was edited.

Line 132

…tissue pieces were plated in 75 cm2 culture flasks for explant culture using a growth medium containing high-glucose Dulbecco’s modified Eagle’s medium (DMEM, Glutamax;…

As suggested by the reviewer this was edited.

Line 260

as seen in figure 2 = FIGURE 3

 As suggested by the reviewer this was edited

line 438

…in the clinical application for prevention or treatment Graft-versus-host disease (GVHD) which…-authors already used this abbreviation in the text.

As suggested by the reviewer this was corrected.

Line 477

Sparrow Hospital Clinical Research Institute

As suggested by the reviewer this was corrected.

Authors are strongly recommended to check other misprints in the text of manuscript.

The authors appreciate the reviewer’s comment.

The text of the manuscript was carefully reviewed and significant review and revision was completed to fix all other misprints.

Reviewer 3 Report

This study highlights the (better) options of maintaining the HSCs ex-vivo derived from placental cord blood. The study rational is good, but the experimental design lacks the scientific depth. 

-Please get some english editing services. And also make sure that the text format should remain the same throughout out the manuscript. (See the Discussion section where fonts are keep changing with the change of paragraph.

-The discussion part lacks the depth and most part is the repetition of result section. Please don’t repeat writing result section entirely in the discussion part. Please refer to more published manuscripts as an example, and see how it should be articulated. 

- Most of the cited literature is old, especially the introduction section. I highly recommend the authors to put the updated citations.

- what different percentages of cell populations exist in Cord Blood that would account for differences in EVs compared to EVs derived from MSCs culture. It should be experimentally shown (what different percentage of cell population exist in cord blood) and should be discussed in the manuscript.

-The characterisation of CBP EVs for proteomics and for coding and non-coding RNA may not necessarily needed in this study but the authors can at least looked into existed MSCs EVs and CBP EVs profile (past manuscript) and can mention some of leading differentiated EV cargos that would be beneficial for HSC growth.

- Here the isolated EVs from placental derived MSCs was cultured with FBS. Fetal bovine serum (FBS), human serum, or human platelet lysate (HPL) are crucial media supplements, but also constitute a major source of EVs and EV-like particles. FBS free medium should have been used instead. This concern can be brought up in the discussion section. And any attempt to extend this work in future with EVs purification should avoid the use of FBS.

Author Response

Reviewer 3 Comments

Comments and Suggestions for Authors

This study highlights the (better) options of maintaining the HSCs ex-vivo derived from placental cord blood. The study rational is good, but the experimental design lacks the scientific depth. 

-Please get some english editing services. And also make sure that the text format should remain the same throughout out the manuscript. (See the Discussion section where fonts are keep changing with the change of paragraph.

As suggested by the reviewer, the manuscript was reviewed by English editing service to review the manuscript and correct any misprint and fix the fonts.

-The discussion part lacks the depth and most part is the repetition of result section. Please don’t repeat writing result section entirely in the discussion part. Please refer to more published manuscripts as an example, and see how it should be articulated. 

This was reviewed and corrected as suggested by the reviewer. The repetition of results was deleted.

- Most of the cited literature is old, especially the introduction section. I highly recommend the authors to put the updated citations.

As suggested by the reviewer, an updated citations were added to the manuscript and older citations were deleted.

- what different percentages of cell populations exist in Cord Blood that would account for differences in EVs compared to EVs derived from MSCs culture. It should be experimentally shown (what different percentage of cell population exist in cord blood) and should be discussed in the manuscript.

The current study was not designed to check the different cell population in the cord blood which may contribute to the EVs in cord blood plasma. Future work will be done to try to identify these cell populations.

-The characterisation of CBP EVs for proteomics and for coding and non-coding RNA may not necessarily needed in this study but the authors can at least looked into existed MSCs EVs and CBP EVs profile (past manuscript) and can mention some of leading differentiated EV cargos that would be beneficial for HSC growth.

As suggested by the reviewer, the section was revised and the following is the revision that was added to the revised manuscript page 28 (line 479-484).

Xie H. and colleagues have reported that BM MSCs derived microvesicles (MVs) were shown to enhance the ex vivo expansion of cord blood CD34+ HSCs and cord blood mononuclear cells with comparable results to MSCs‐HSCs coculture system [33]. In addition, they reported the genomic analyses of adult BM MSCs-MVs and found multiple miRNAs that are involved in the regulation of Wnt/?-catenin signaling pathway which is crucial for the regulation of hematopoiesis, promoting self-renewal and inhibiting HSCs differentiation.

- Here the isolated EVs from placental derived MSCs was cultured with FBS. Fetal bovine serum (FBS), human serum, or human platelet lysate (HPL) are crucial media supplements, but also constitute a major source of EVs and EV-like particles. FBS free medium should have been used instead. This concern can be brought up in the discussion section. And any attempt to extend this work in future with EVs purification should avoid the use of FBS.

The authors thank the reviewer for this important comment.

The details about the use of EV depleted media have been in the methods section in page 9 (line170-174) as below:

EV-depleted serum was prepared by 18-hour ultracentrifugation at 100,000 x g at 4° C [46]. For each of passages 3-5, MSCs were seeded at a density of 1x104 cells/cm2 in complete growth medium for 24 hours and replaced by an EV-depleted medium to obtain the culture supernatant of MSCs (DMEM-high glucose supplemented with 10% EV-depleted serum and 1% antibiotic antimycotic). The conditioned medium was collected after 48 hours for isolation of exosomes.

In addition, figure 1 legend in page 7-8 (line142-145)  was revised for more clarification as follow:

… expansion of MSCs via subculture from primary MSCs to passage 3-5, replacement of the culture media with EV depleted media when the expanded cells were at 60-80% confluence, then collection of the MSCs culture supernatant after 48 hours and exosomes isolation from CBP and MSCs culture supernatants.

Round 2

Reviewer 3 Report

Accept in the current form